# Is the Efficiency Score an Indicator for Incident Hypertension in the Community Population of Western China?

**DOI:** 10.3390/ijerph181910132

**Published:** 2021-09-27

**Authors:** Yangwen Yu, Yun Chen, Yiying Wang, Lisha Yu, Tao Liu, Chaowei Fu

**Affiliations:** 1Guizhou Center for Disease Control and Prevention, Guiyang 550004, China; yuyangweny@163.com (Y.Y.); wyy123789123789@163.com (Y.W.); lishayu621@163.com (L.Y.); 2School of Public Health & Key Laboratory of Public Health Safety & NHC Key Laboratory of Health Technology Assessment, Fudan University, Shanghai 200032, China; 18211020001@fudan.edu.cn

**Keywords:** hypertension, efficiency score, cohort study

## Abstract

We aimed to explore the association between the efficiency score and the risk of hypertension. We conducted a prospective cohort study of 2412 adults aged 40 years or above without hypertension in Guizhou, China from 2010 to 2020. The data envelopment analysis input-oriented DEA-CCR model was used to calculate the efficiency scores. The Cox regression model was used to assess the relationship between the efficiency score and incident hypertension. The dose–response relationship was evaluated by restricted cubic spline. Quantile regression was used to analyze the effect of efficiency scores on SBP and DBP. A total of 857 new hypertension cases were identified with a mean follow-up of 6.88 years. The efficiency score was lower in the new hypertension cases than participants without hypertension (0.70 vs. 0.67). After adjusting for possible confounding factors, the HR of hypertension risk was 0.20 (95%CI: 0.09, 0.42) for per 0.1 increase in the efficiency score. The dose–response relationship showed a non-linear relationship between the efficiency score and hypertension risk. Our results showed that the efficiency score was a cost-effective tool to identify those at a high risk of hypertension, and suggested targeted preventive measures should be undertaken.

## 1. Introduction

Every year, about 17 million deaths worldwide are caused by cardiovascular diseases, of which 9.4 million by the complications of hypertension [1]. Hypertension is the third major risk factor for high-burden diseases, resulting in 1.04 million deaths and 21.21 million disability-adjusted life years loss [2], placing a heavy economic burden on individuals, families, and society. The World Health Organization lists high blood pressure as one of the main public health challenges. With low awareness, treatment, and control rate, a survey showed that one in four was found hypertension in China [3]. The prevalence rate of hypertension in Guizhou Province reached 29.2%, and the incidence of hypertension is on the rise at a younger age [4]. The prevalence of prehypertension in people with normal blood pressure ranges from 25% to 50% [5]. Due to the slow onset process and complex risk factors of hypertension, finding effective methods to assess the risk of hypertension and conducting reasonable intervention in advance can reduce the occurrence of hypertension events effectively [6].

At present, there are two methods to identify the risk of hypertension at the individual level. One is genetic testing, which is difficult to implement in the general population. Another method is using socioeconomic status indicators to identify high-risk groups [7], but there is no widely accepted socioeconomic status indicator at present. Therefore, the exploration of widely accepted indicators of socioeconomic status is an important topic.

Studies have shown that data envelopment analysis-based efficiency scores predict the onset of hypertension with the same validity as traditional predictors [8]. Based on the Guizhou Population Health Cohort Study (GPHCS), an efficiency score was applied to identify the high risk of hypertension, providing a basis for early hypertension risk in intervention in this study.

## 2. Materials and Methods

### 2.1. Study Design and Population

This Guizhou Population Health Cohort Study (GPHCS) was established in Guizhou Province in 2010–2012. Using the stratified cluster sampling method, a total of 9280 permanent residents aged 18 years and above were selected from 12 counties in Guizhou Province in the baseline survey. The information was collected by trained investigators using a family and individual questionnaire via a face-to-face interview. This study was approved by the Ethics Committee of the center of disease control and prevention of Guizhou Province.

Baseline information, including demographic characteristics (sex, age), lifestyle (smoking status, alcohol use, physical activity, and diet), and history of chronic diseases. The physical examination, including height, weight, and blood pressure were measured by trained investigators. The salt intake of the whole family was collected by the question “How much salt do your family usually eat a month?”, and then the daily salt intake per capita was calculated. A food frequency questionnaire (FFQ) was used to investigate the grain and cereal intake, meat intake, and vegetable and fruit intake. Blood pressure was measured using a mercury sphygmomanometer on the right hand in a sitting position three times at 5-min intervals to confirm blood pressure values and ensure controls are normotensive. The average of the three readings each taken at 5-min intervals was considered as the final reading. Venous blood samples were obtained from participants after at least 8h overnight fast to measure fasting plasma glucose (FPG), total cholesterol (TC), triglycerides (TG), high-density lipoprotein cholesterol (HDL-C), and low-density lipoprotein cholesterol (LDL-C).

A total of 8163 participants were followed up from 2016 to 2020, and 1117 (12.04%) were lost. We further excluded 2119 individuals with a history of hypertension at baseline. A total of 2412 subjects aged 40 years old and over were included in the final analysis. 

### 2.2. Definitions

Diagnostic criteria of hypertension: systolic blood pressure ≥ 140 mmHg and/or diastolic blood pressure ≥ 90 mmHg, or previously diagnosed as hypertension [9]. Dyslipidemia was defined as TC ≥ 6.22 mmol/L, or LDL-C ≥ 4.14 mmol/L, or HDL-C < 1.04 mmol/L, or TG ≥ 2.26 mmol/L, and/or receiving treatment for dyslipidemia [10]. Current smoking refers to smoking tobacco products including manufactured or locally produced within the last 30 days. Abuse drinking refers to the drinking behavior that the average daily pure alcohol intake of male drinkers is ≥61 g, and that of female drinkers is ≥41 g; not abusing drinking refers to the drinking behavior that the average daily pure alcohol intake of male drinkers is ≥41 g and <61 g, and that of female drinkers is ≥21 g and <41 g [11]. Activity: The physical activity information was collected by IPAQ questionnaire, measured by metabolic equivalent (MET), it refers to the ratio of metabolic rate when people are active and metabolic rate when they are at rest, which is usually used to express the intensity of physical activity [12]. The values continue to be used for the analysis of IPAQ data: Moderate PA = 4.0 METs and Vigorous PA = 8.0 METs [13]. Body mass index (BMI) is defined as the individual’s body mass in kilograms divided by the height in meters squared.

### 2.3. DEA Analysis

Efficiency refers to the result generated by the budget input of an institution or organization within a given period, which can be understood as the proportional relationship between input and output [14]. Data envelopment analysis (DEA) is a commonly used method for efficiency evaluation and analysis. It is a method to evaluate the efficiency of a unit with the same type by taking into multiple input indexes and multiple-output indexes in the form of linear programming. When evaluating similar departments or units, the evaluation is usually based on their “input” and “output” data. DEA method is a powerful tool to deal with this kind of problem. After a comprehensive analysis of the input and output data of the decision-making unit group through a data programming model, it can get the quantitative index of the comprehensive efficiency of each decision-making unit (DMU) relative to other units. At present, data envelopment analysis is widely used in service efficiency evaluation at home and abroad [15,16]. 

“Charnes, Cooper, and Rhodes model (CCR) is the original and most commonly used DEA model. The formula of the basic CCR model is as follows [17]:$$h_{k}=max\overset{s}{\underset{r=1}{\sum }}u_{r}y_{rk}$$
Subject to:
$$\overset{m}{\underset{i=1}{\sum }}v_{i}x_{ij}-\overset{s}{\underset{r=1}{\sum }}u_{r}y_{rj}\geq 0\;\;\mathrm{for}\;j\;=\;1,\ldots ,\;n\;\overset{m}{\underset{i=1}{\sum }}v_{i}x_{ik}=1\;\;u_{r}\geq 0\;\mathrm{for}\;r=1,\ldots ,\;s\;\;u_{i}\geq 0\;\mathrm{for}\;i=1,\ldots ,\;m$$
if its efficiency value is equal to one and its input excesses and output short falls are zero. CCR model includes input-oriented model (I model) and output-oriented model (O model). I model means that the output is constant and the demand input is minimum; O model means that the input is certain and the maximum output is sought.”

In this study, an input-oriented DEA-CCR model was applied to measure the comprehensive efficiency score of each object. The reciprocal of baseline salt intake, total activity, grain and cereal intake, meat intake, and baseline BMI were used as input indexes, and baseline systolic blood pressure (SBP) and diastolic blood pressure (DBP) were used as output indexes to calculate the efficiency score of each individual.

### 2.4. Statistical Analysis

Continuous variables were described as $\bar{x}+s$ and compared by the Student’s t-test. Categorical variables were described as frequency (percentage) and compared by the χ^2^ test. The association between the efficiency score and risk of incident hypertension was estimated by Cox proportional risk regression model: (1) Model 1: crude model; (2) Model 2: adjusted for age and gender; (3) Model 3: further adjusted smoking, alcohol use, daily salt intake, vegetable and fruit intake, BMI, SBP, DBP, dyslipidemia based on Model 2. The dose–response relationship between the efficiency score and hypertension risk was assessed by the restricted cubic spline, with nodes set at the 25th, 50th, 75th, and 95th percentiles of the efficiency score. Quantile regression (QR) was used to analyze the relationship between efficiency score and systolic and diastolic blood pressure. All statistical analyses were performed in MAXDEA2.0 and STATA16.0. All statistical tests were two-sided and *p* < 0.05 was considered statistically significant.

## 3. Results

### 3.1. Baseline Characteristics

Among the 2412 participants included in this analysis, 43.8% were male, with a mean age of 51.83 ± 9.15 years old. During the mean follow-up years of 6.88, 857 new cases of hypertension were identified. Compared with the non-hypertension, the new cases were more likely to be male, smokers, with higher age, SBP, and DBP. The average daily intakes of salt, meat, and vegetable and fruit intake were relatively low in new hypertension participants. The mean efficiency score of the subjects was 0.67 ± 0.11 (min: 0.48, Max: 1.00), and the hypertension group (0.67 ± 0.11) was significantly lower than the non-hypertension group (0.70 ± 0.12). More details are presented in Table 1.

### 3.2. The Association between the Efficiency Score and Risk of Incident Hypertension

In Table 2, the results showed that the efficiency score was associated with a decreased risk of hypertension with an HR of 0.34 (95%CI: 0.18, 0.63) for per 0.1 increase, and the adjusted HR increased slightly after being adjusted for age and gender. After further adjusted for smoking, alcohol use, daily salt intake, vegetable and fruit intake, BMI, SBP, DBP, dyslipidemia, efficiency score was still significantly associated with the risk of hypertension, with the HR of 0.20 (95% CI: 0.09, 0.42). In subgroup analysis, the adjusted HRs of the efficiency score were lower in males or participants over 60 years older than females or those under 60 years old.

Restricted cubic spline was used to estimate the potential non-linear relationship, and showed that the risk of hypertension decreased with the increase in efficiency scores (*p* < 0.001). The dose–response curve was steep at first and then smooth (Figure 1).

### 3.3. Quantile Regression between Efficiency Score and Systolic and Diastolic Blood Pressure in Follow-Up

The coefficient estimates and 95%CI for the associations of the efficiency score with SBP and DBP across the quantile levels of the efficiency score are given in Table 3. In the QR analysis, the efficiency score was significantly associated with SBP after adjusting potential confounding factors, except the quantile level of 10% and 90%. The estimates were between −24.83 to −13.39 among the quantile of 20% to 80%. The efficiency score influenced SBP between 117 to 143 mmHg (quantile SBP level of 20% to 80%). However, the association between the efficiency score and DBP was found to be not significant except for the quantile level of 60%, 80%, and 90%, of which DBP was between 80 to 92 mmHg.

## 4. Discussion

In this study, the efficiency score calculated by the DEA was 0.67 in the hypertension group and was lower than that in the non-hypertension group (0.70). After controlling for traditional risk factors, we found that the higher the efficiency score was associated with a lower risk of incident hypertension, and the protective effect was stronger in males and individuals aged 60 years or older. The result of the dose–response reaction suggested a non-linear relationship for the efficiency score with the risk of hypertension, showing a negative correlation.

The result of the QR analysis indicated that the efficiency score affects SBP and DBP. The effect on SBP was mainly between 117 and 143 mmHg. The effects on DBP were mainly between 80 and 93mmHg. The efficiency score was sensitive to the upper limit of normal SBP and DBP. In other words, the efficiency score calculated by the DEA was enable used to provide primary preventive intervention. The result was consistent with the literature report that the efficiency score could not predict the onset of hypertension in the extreme-risk group [8]. 

The efficiency score calculated based on data envelopment analysis is already being used for individual prevention [18]. The efficiency score [8] calculated by data envelopment analysis was compared with traditional factors to identify people at risk of hypertension, showing that efficiency score and traditional factors can be complementary, especially in the population without conventional risk factors, this indicates that the application of the efficiency score might be more efficient for the primary prevention of hypertension, in a population with no conventional risk factor.

At present, disease risk prediction is mostly based on traditional risk factors, and appropriate statistical analysis models [19] and artificial intelligence [20] are adopted to construct the disease risk prediction model to predict the occurrence probability of the disease, the traditional factors include age, parental hypertension, SBP, DBP, BMI. However, the range of high-risk groups identified for traditional risk factors is too large, especially in the western region where the medical and health resources are relatively scarce. This is based on the presence or absence of traditional influencing factors, which is not conducive to realize the advance of the prevention and treatment of diseases and lacks targeted intervention guidance for individuals found. The hypertension risk prediction based on efficiency score can aim at the efficiency frontier and directly calculate the improvement direction and target value for individual indicators, to improve the efficiency score and reduce the risk of disease.

In this research, based on previous studies, the environmental factors and blood pressure were analyzed to calculate the correlation between the efficiency score and the incidence of future hypertension, and five easily measured input indicators and two output indicators of systolic blood pressure and diastolic blood pressure were selected, and efficiency scores were calculated according to the data envelopment analysis DEA-CCR model, which innovated the screening method for early identification of people at risk of hypertension.

There were several strengths in this study. First, to the best of our knowledge, it is the first cohort study developing an efficiency score based on the DEA-CCR model and estimating the association between the efficiency score and the risk of incident hypertension. Second, a long follow-up period with a relatively low loss to follow-up rate limited the potential bias for association estimates. Additionally, potential limitations of the study should be considered. First, the sample size of this study was small. Second, owing to the fact that this study was limited to individuals over 40 years old, our findings should be generalizable to all people, with caution. With the advanced age of onset of hypertension, it is also imperative to identify individuals under 40 years old who are at a high risk of hypertension. A future large cohort study for all age groups is needed. Third, during the follow-up, there was no collection of whether hypertension was secondary to other diseases or whether drugs leading to elevated blood pressure were taken; this needs to be improved in future studies.

## 5. Conclusions

The efficiency score calculated by the DEA-CCR model was associated with the risk of incident hypertension. Our results provide new evidence that the efficiency score is a cost-effective tool to identify those at a high risk of hypertension among healthy individuals, and suggest that targeted preventive measures should be undertaken to prevent the occurrence of hypertension in those at risk.

## Figures and Tables

**Figure 1 ijerph-18-10132-f001:**
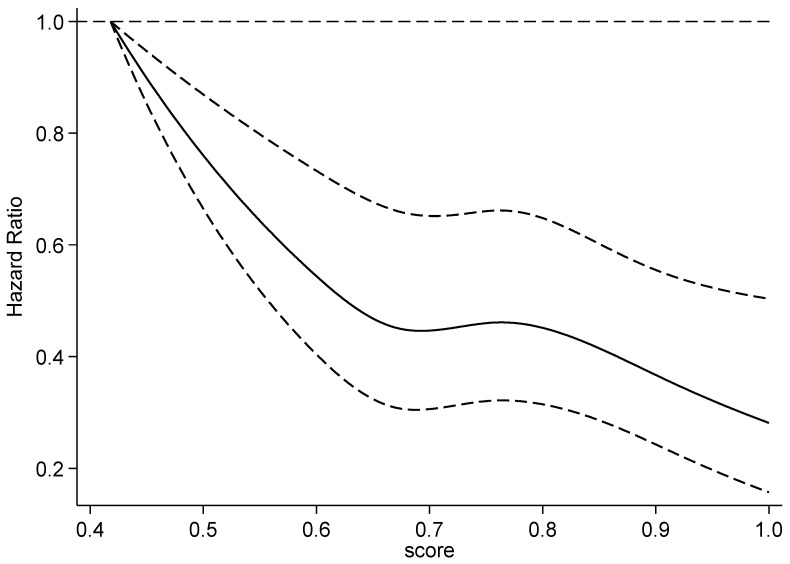
Dose–response relationship between the efficiency score and risk of incident hypertension.

**Table 1 ijerph-18-10132-t001:** Baseline characteristics of the study population.

Characteristics	Total (*n* = 2412)	Hypertension (*n* = 857)	Non-Hypertension (*n* = 1555)	*t/Χ^2^* Value	*p* Value
Gender				11.070	0.001
Male	1056 (43.8)	414 (48.3)	642 (41.3)		
Female	1356 (56.2)	443 (51.7)	913 (58.7)		
Age	51.83 ± 9.15	52.38 ± 9.46	51.52 ± 8.96	2.193	0.028
Smoking				4.012	0.045
Yes	614 (26.8)	234 (29.3)	380 (25.4)		
No	1681 (73.2)	565 (70.7)	1116 (74.6)		
Alcohol use				4.127	0.127
Yes, abuse	63 (2.6)	30 (10.1)	33 (6.3)		
Yes, non-abuse	756 (31.3)	266 (89.9)	490 (93.7)		
No	1593 (66.0)	561 (65.5)	1032 (66.4)		
METs	7655.02 ± 157.49	7452.45 ± 256.73	7766.68 ± 199.15	0.955	0.340
Salt intake (g/d)	11.98 ± 0.16	11.35 ± 0.21	12.32 ± 0.21	2.986	0.003
Grain and cereal intake (g/d)	428.45 ± 4.58	430.47 ± 9.08	427.33 ± 5.03	0.328	0.743
Meat intake (g/d)	93.38 ± 2.12	82.37 ± 3.27	99.44 ± 2.74	3.870	<0.001
Vegetable and fruit intake (g/d)	397.87 ± 4.61	384.79 ± 208.47	405.09 ± 235.11	2.112	0.035
BMI (kg/m^2^)	22.74 ± 0.06	22.62 ± 0.10	22.81 ± 0.08	1.469	0.142
SBP (mmHg)	118.80 ± 11.51	120.64 ± 11.04	117.81 ± 11.64	5.771	<0.001
DBP (mmHg)	74.60 ± 7.80	75.17 ± 7.54	74.29 ± 7.92	2.650	0.008
TC	4.82 ± 1.29	4.81 ± 1.22	4.82 ± 1.34	0.220	0.826
TG	1.76 ± 1.78	1.76 ± 1.61	1.75 ± 1.86	0.172	0.864
HDL-C	1.46 ± 0.59	1.47 ± 0.57	1.46 ± 0.59	0.314	0.754
LDL-C	2.68 ± 1.14	2.65 ± 1.14	2.69 ± 1.15	0.902	0.367
Efficiency score	0.69 ± 0.11	0.67 ± 0.11	0.70 ± 0.12	6.810	<0.001

Abbreviations: METs, metabolic equivalents; BMI, body mass index; SBP, systolic blood pressure; DBP, diastolic blood pressure; TC, total cholesterol; TG, triglycerides; HDL-C, high-density lipoprotein cholesterol; LDL-C, low-density lipoprotein cholesterol.

**Table 2 ijerph-18-10132-t002:** Association between the efficiency score and risk of incident hypertension.

	Hazard Ratio (95% Confidence Interval)		
	Model 1	Model 2	Model 3
Efficiency score (per 0.1 increase)	0.34 (0.18, 0.63) **	0.36 (0.19, 0.66) **	0.20 (0.09, 0.42) *
Subgroup analysis			
Gender			
Male	0.22 (0.09, 0.55) **	0.23 (0.09, 0.56) **	0.15 (0.05, 0.44) **
Female	0.46 (0.20, 1.07)	0.54 (0.23, 1.24)	0.28 (0.10, 0.78) **
Age, years			
<60	0.49 (0.24, 1.00)	0.47 (0.23, 0.95) *	0.26 (0.11, 0.62) **
≥60	0.16 (0.05, 0.57) **	0.16 (0.05, 0.57) **	0.07 (0.01, 0.35) **

**Note:** Model 1, basic model; Model 2, adjusted for age and gender; Model 3, Model 2 plus smoking, alcohol use, daily salt intake, vegetable and fruit intake, BMI, SBP, DBP, dyslipidemia. ** *p* < 0.01; * *p* < 0.05. Abbreviations: Hazard Ratio (HR); 95% confidence interval (95% CI).

**Table 3 ijerph-18-10132-t003:** Quantile regression between the efficiency score with systolic and diastolic blood pressure.

Percentile of Efficiency Score	Coefficient of SBP(95%CI)	Coefficient of DBP(95%CI)
10	−17.88 (−37.45, 1.687)	0.03 (−9.94, 9.99)
20	−13.39 (−26.61, −0.17) *	−3.39 (−15.90, 9.12)
30	−13.91 (−22.93, −4.90) **	−2.84 (−14.67, 8.99)
40	−16.81 (−28.43, −5.19) **	−4.19 (−13.35, 4.98)
50	−19.75 (−32.09, −7.42) **	−4.00 (−10.85, 2.86)
60	−21.88 (−38.54, −5.21) *	−7.36 (−14.14, −0.58) *
70	−24.83 (−44.69, −4.96) *	−6.70 (−15.85, 2.44)
80	−19.63 (−34.33, −4.93) **	−11.35 (−19.99, −2.71) *
90	−30.92 (−65.79, 3.95)	−16.50 (−27.54, −5.45) **

Note: Adjusted for age, gender, smoking, alcohol use, daily salt intake, vegetable and fruit intake, BMI, dyslipidemia. ** *p* < 0.01; * *p* < 0.05. Abbreviations: SBP, systolic blood pressure; DBP, diastolic blood pressure.

## Data Availability

The datasets generated for this study are available on request to the corresponding author.

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
