# Peer review of "Is the Efficiency Score an Indicator for Incident Hypertension in the Community Population of Western China?"

_ijerph, 2021, doi:10.3390/ijerph181910132_

Round 1
Reviewer 1 Report
Is the efficiency score an indicator for incident hypertension in the community population of Western China?
The publication has a social and scientific value because it proposes a nonparametric tool for the analysis of hypertension risk factors to be used in the area of prevention.
Mathematical algorithms are bound to change as more information becomes available. Hypertension as other chronic diseases currently have to be observed from a life cycle approach because its origin has to do from gestation (genetics), the period of childhood, adolescence, young adulthood and adults. The interventions to be made at the beginning of our cycle are not the same as those that can be used in adulthood. Today, emphasis is placed on early intervention, therefore prevention has more value in the early stages of our lives and would have a greater impact on reducing the incidence of these diseases. In adulthood they can only be controlled.
I have some doubts:
It is my understanding that the people who participated in the study at baseline had no signs or symptoms of hypertension. However, within the people who developed hypertension in the follow-up period it was established whether it was of primary origin (essential hypertension) or secondary to underlying diseases.
It is not mentioned whether the interview asked about the consumption of some medications such as contraceptives, anti-influenza, decongestants, over-the-counter analgesics and some prescription drugs. As well as illicit drugs, such as cocaine and amphetamines. These consumptions can also raise blood pressure.
Regarding gender, the literature establishes that hypertension is more frequent in men. However, this changes with age. As women enter the menopause period, especially in their early 50's, hypertension becomes more frequent in women than in men.
Best regards
Author Response
Response to Reviewer 1 Comments
The publication has a social and scientific value because it proposes a nonparametric tool for the analysis of hypertension risk factors to be used in the area of prevention.
Mathematical algorithms are bound to change as more information becomes available. Hypertension as other chronic diseases currently have to be observed from a life cycle approach because its origin has to do from gestation (genetics), the period of childhood, adolescence, young adulthood and adults. The interventions to be made at the beginning of our cycle are not the same as those that can be used in adulthood. Today, emphasis is placed on early intervention, therefore prevention has more value in the early stages of our lives and would have a greater impact on reducing the incidence of these diseases. In adulthood they can only be controlled.
I have some doubts:
Point 1: It is my understanding that the people who participated in the study at baseline had no signs or symptoms of hypertension. However, within the people who developed hypertension in the follow-up period it was established whether it was of primary origin (essential hypertension) or secondary to underlying diseases.
Response 1: Thanks! The hypertension is defined as: systolic blood pressure ≥140mmHg and/or diastolic blood pressure ≥90 mmHg, or previously diagnosed as hypertension. We are unable to distinguish between primary or secondary hypertension with the follow-up data. A future better designed study is needed. It has been stated as a limitation in the Discussion section.
“Third, during the follow-up, there was no collection of whether hypertension was secondary to other diseases or whether drugs leading to elevated blood pressure were taken, which need to be improved in future studies.” (Discussion section, lines 239-242, page 7)
Point 2: It is not mentioned whether the interview asked about the consumption of some medications such as contraceptives, anti-influenza, decongestants, over-the-counter analgesics and some prescription drugs. As well as illicit drugs, such as cocaine and amphetamines. These consumptions can also raise blood pressure.
Response 2: We agree. However, there is no data about the consumption of the drugs mentioned above based on the cohort, and it will be considered in future follow-up visit. More discussion has added at the limitation section.
“Third, during the follow-up, there was no collection of whether hypertension was secondary to other diseases or whether drugs leading to elevated blood pressure were taken, which need to be improved in future studies.” (Discussion section, lines 239-242, page 7)
Point 3: Regarding gender, the literature establishes that hypertension is more frequent in men. However, this changes with age. As women enter the menopause period, especially in their early 50's, hypertension becomes more frequent in women than in men.
Response 3: Gender and age are unalterable influencing factors, which are adjusted in the current analysis. This study recruited people aged 40 years and above, with an average age of 51.83 ± 9.15 years old. In the subgroup analysis, the adjusted HRs of the efficiency score were lower in males or participants over 60 years older than females or those under 60 years old.
Reviewer 2 Report
The advantages of the work submitted for review are: current and very important issues, long-term and multi-directional research, a sufficiently large study population and proper analysis of the obtained results. The paper is well written and gives a good overview of the issue. Therefore, I think that it can be accepted for publication.
Author Response
Response to Reviewer 2 Comments
The advantages of the work submitted for review are: current and very important issues, long-term and multi-directional research, a sufficiently large study population and proper analysis of the obtained results. The paper is well written and gives a good overview of the issue. Therefore, I think that it can be accepted for publication.
Response: Thanks for your comments.
Reviewer 3 Report
- The article uses the abbreviation DEA-CCR. It explains what DEA (Data Envelopment Analysis) is, but nowhere does it say what CCR means.
- Line 127 Change "More details were presented in Table1." for "More details are presented in Table 1."
- The authors indicate that they have measured physical activity in METS, but they do not explain the measurement system. Did they use accelerometers, questionnaires such as the Paffenbarger or the IPAQ? They should explain how they have obtained the information on physical activity.
- The paper indicates that Salt intake, Grain and cereal intake (g/d), Meat intake, and Vegetable and fruit intake (g/d) are measured. There is no information on how these variables have been obtained, e.g., 24-hour recall questionnaire, food frequency questionnaire, or another method. They should indicate the procedure for obtaining the information.
- I do not understand Table 2. It is indicated that the Hazard Ratio is proportional, but if the data for men and women are provided, one of the groups would be the baseline and have an HR of 1. There must be some error.
